# Supervisory Career Support and Workplace Wellbeing in Chinese Healthcare Workers: The Mediating Role of Career Commitment and the Moderating Role of Future Work Self-Salience

Guangyi Xu , Zhen Li * and Hongli Wang *

School of Business Administration, South China University of Technology, Guangzhou 510640, China; xugy1990@163.com
* Correspondence: lee_lizhen2020@163.com (Z.L.); bmhlwang@scut.edu.cn (H.W.)

**Abstract:** In the context of the sustainability goals of organizations, there is a dilemma regarding enhancing healthcare workers' career commitment and wellbeing, especially during the COVID-19 crisis. This study focuses on the underlying mechanism in the relationship between supervisory career support and employee wellbeing. Drawing upon the career motivation perspective, we investigate the mediating role of career commitment and moderating effect of future work self-salience (FWSS) in this relationship. Two-wave data were collected from a sample of 213 full-time healthcare workers from three public hospitals located in Southern China. Results in this study revealed that supervisory career support influences career commitment in health workers, which in turn enhances their wellbeing at the workplace. Moreover, the effect of supervisory career support on career commitment was found to be stronger for individuals with low FWSS compared to those with high FWSS. These findings also enlighten us on how to enhance employees' career commitment and workplace wellbeing.

**Keywords:** supervisory career support; career commitment; future work self-salience; workplace wellbeing; healthcare workers

## 1. Introduction

Organizational sustainability has drawn more and more attention for the last two decades [1–3]. An increasing number of organizations regard sustainability as part of their goal and mission. Out of the three basic dimensions of organizational sustainability, however, social dimension is not attributed enough importance by academia compared to the economic and environmental dimensions [4–6]. The social performance of sustainable organizations is the human dimension, which is closely related to employees' wellbeing [6,7]. Employee wellbeing is a particularly important issue for sustainable organizations because it is a key driver of employees' affective commitment, job performance, and organizational citizenship behavior [8–10]. However, during the COVID-19 crisis, healthcare professionals such as doctors and nurses are facing unprecedented challenges in terms of their wellbeing, which is affected by exhaustion due to heavy workloads and the fear of becoming infected and infecting others [11]. Healthcare professionals with a higher level of wellbeing are more likely to be enthusiastic about their job and provide patient-centered care [12]. Therefore, healthcare workers' wellbeing has attracted close attention from scholars and managers [13,14].

As employee wellbeing is one of the sustainability-related goals of most organizations, it is necessary to investigate how to enhance healthcare workers' wellbeing during the COVID-19 pandemic, which is of great significance for organizations to continue to meet their sustainability goals in the face of an external crisis [15,16]. Since leaders are generally viewed as agents of organizations, leadership styles and behaviors will have a great influence on their subordinates' emotional experience and psychological status and

affect their wellbeing [17,18]. Although the existing research has examined the impact of supervisory support on healthcare workers' wellbeing [19,20], we know relatively little about the effects of career support variables on workplace wellbeing. Past research has linked perceived career support to attitudinal variables such as affective commitment [21], intention to leave [22], and work engagement [23]. Therefore, the first purpose of this research was to examine how supervisory career support, i.e., the professional support an employee receives directly from their supervisor [24], affects healthcare workers' wellbeing at the workplace.

While it is important to know what can predict workplace wellbeing, knowing how and when predictors work can help us to advance even further. So far, little is known about how employees' wellbeing in the workplace is determined [17]. For example, through what intervention mechanisms is supervisory career support associated with workplace wellbeing? Therefore, this study attempted to propose career commitment, which represents the career-based identities of employees, as a potential intervention variable to address the question of why supervisory career support can predict employees' wellbeing at the workplace. Career commitment is believed to mediate the relationship between supportive work conditions and employees' wellbeing, and previous studies have found that career commitment can mediate the relationship between professional identity and subjective wellbeing [25]. Therefore, we expect it to play a similar role in the link between supervisory career support and workplace wellbeing. Accordingly, the second purpose of this study was to investigate the mediating effect of career commitment. In addition, the theory of career motivation [26,27] suggests that the effect of situational characteristics on career decision making and career behavior is influenced by individual characteristics. Thus, this study further explored the moderating role of future work self-salience (FWSS), which represents the ease of construction and clarity of an individual's hoped-for work-based identity [28]. Namely, we demonstrate the individual factor (e.g., FWSS) that makes supervisory career support more or less effective.

The contribution of this study mainly includes the following aspects. First, unlike previous research that focused on perceived organizational support, supervisory support, or coworker support for prediction of employees' wellbeing, this study also examines the less-studied support variable of supervisory career support as a predictor of employees' wellbeing at the workplace. From the perspective of career support, we also offer a new idea regarding how to enhance healthcare workers' wellbeing in the face of external crises. Secondly, in previous studies, researchers focused on the direct impact of supporting variables on employees' wellbeing. This study further integrates insights from the career motivation literature to examine the mediation model, which links supervisory career support and workplace wellbeing through the mediating mechanism of career commitment, which represents the career-based identities of employees. That is, this study highlights the significance of supervisory support to enhancing healthcare workers' career commitment. Finally, this study further examines the moderating effect of personality on the relationship between supervisory career support and career commitment. By examining the moderating role of FWSS, this study aims to prove that the influence of supervisory career support on career commitment is not universal among individuals but depends on the significance of their expected job-related identity.

## 2. Theoretical Backgrounds and Hypotheses

In order to better understand the influence of perceived career support on workplace wellbeing, we draw from the theory of career motivation [26,27], which has proved useful in understanding employees' career decisions and behaviors [29–31]. According to the theory of career motivation, personal and situational characteristics are often considered determinants of career decisions and behaviors. Career motivation is usually conceptualized as a multidimensional structure within an individual, which is influenced by the situation and reflected in their career decisions and behaviors [32]. The individual characteristic dimensions are needs, interests, and personality variables that are potentially related to a person's

career. These dimensions can be divided into three domains: career identity, career insight, and career resilience [26]. Career identity refers to the extent to which individuals define themselves according to their career; career resilience means that an individual sticks to their career in an unsatisfactory environment; and career insight refers to whether people have a realistic perception of themselves, their organization, and their career [26]. In these three dimensions, career insight is conceptually similar to FWSS, while career identity and career resilience are similar to career commitment [30]. In addition, many situational factors in the work environment, such as leadership style, job design, and career development plans, may also be important for career motivation [26]. Supervisory career support, thus, is generally viewed as a favorable situational characteristic for career development.

Based on the theory of career motivation, we posit that when healthcare workers are offered career support by their supervisors, it signals that supervisors care about their professional needs and development and enhance their workplace wellbeing. Furthermore, supervisory career support reflects positive feedback to team members to a certain extent and can promote the establishment of emotional connections between an individual and their career. When career commitment is conceptually translated into the intensity of an individual's motivation to work in a chosen career role, career motivation can be used as a framework to examine how supervisory career support affects an employee's workplace wellbeing. In addition, the effect of situational characteristics on career decision making and career behavior is influenced by individual characteristics [25,26]. Therefore, the effect of supervisory career support on career commitment may be influenced by FWSS, which represents the ease of construction and clarity of an individual's hoped-for work-based identity. From a career motivation perspective, we can obtain a coherent picture of the positive influences of supervisory support behavior on healthcare workers.

### 2.1. Supervisory Career Support and Workplace Wellbeing

Supervisory career support refers to the professional support that an employee receives directly from their supervisor [24]. This support can take the form of career guidance, challenging tasks, and performance feedback to promote career development [33]. This study focuses on employees' perception of career support because it is employees' perception of their job that determines their response to career support [23]. Perceived career support is an indispensable resource for employees, which helps them to form a positive mental state [34]. Furthermore, a supportive environment signals to employees that they are valued [23], and employees are likely to form positive expectations about their career development [35].

There are two main philosophical views on employee wellbeing: hedonism and eudaemonism [36]. The former defines wellbeing as the subjective experience of happiness, while the latter regards wellbeing as the result of personal achievement and self-actualization [36]. The validity of the two different paradigms has been accepted by most research on wellbeing [37]. Research on employee wellbeing originates from the consideration of general wellbeing [38], such as subjective wellbeing [39] and psychological wellbeing [40]. However, many previous studies [37,41] have shown that overall wellbeing does not fully reflect wellbeing at work. Some scholars [42] suggest that it is necessary to measure wellbeing according to the specific context to capture the complexity of employees' cognitive and emotional experience at work. In addition, Page and Vella-Brodrick [43] believe that the combination of wellbeing at work and general wellbeing can more accurately assess employees' wellbeing. They proposed a theoretical model of employee wellbeing which comprises three components: subjective wellbeing, psychological wellbeing, and workplace wellbeing. This study mainly explores the impact of supervisory career support on the sense of wellbeing at the workplace.

Although there is no direct evidence that supervisory career support can predict workplace wellbeing, some researchers have found that perceived career support can predict turnover intention [22] and job engagement [23]. According to the theory of career motivation, the practice of supervisory management can be regarded as the signal that

the organization transmits value and expectation to employees [44]. Specifically, supervisory career support, such as career guidance, performance feedback, and challenging work tasks, provides employees with tangible wealth and social emotional resources, which can improve their ability, enthusiasm, and participation opportunities [45]. Thus, supervisory career support signals to subordinates that their supervisors care about their professional needs and employee wellbeing [46]. Employees who perceive career support are likely to see it as a favorable working environment. This enables employees to experience positive work events, such as opportunities to participate in management, training, and promotion, thereby enhancing their workplace wellbeing [47,48]. Therefore, based on these theoretical arguments and empirical evidence, we propose that:

**Hypothesis 1.** *Supervisory career support is positively related to workplace wellbeing.*

### 2.2. The Mediating Effect of Career Commitment

Why should supervisory career support be related to workplace wellbeing? One possible linking variable to explain this relationship is career commitment—the psychological connection between individuals and their profession based on their emotional response to the occupation [49]. This form of commitment highlights an individual's enthusiasm for their chosen occupation [50]. Chang [51] suggested that personal professional attitude can effectively understand how an organization affects employees. As a result, people who are highly engaged in their careers have higher expectations of their employers, and the realization of self-worth will affect their wellbeing. When career commitment is conceptually translated into the intensity of an individual's motivation to work in a chosen role, career motivation can be used as a framework to examine how supervisory career support affects an employee's workplace wellbeing.

There are theoretical and empirical reasons for the mediating effect of expected career commitment on the relationship between supervisory career support and workplace wellbeing. Theoretically, the theory of career motivation [26] provides a foundation for the relationship between supervisory career support and career commitment. According to the theory of career motivation, situational characteristics are often considered important determinants of career decisions and behaviors. Supervisory career support is a good situational condition which reflects organizational opportunities for positive feedback and career advancement to members to a certain extent and can promote the establishment of emotional connections between individuals and their career [29]. Therefore, perceived career support stimulates a sense of responsibility and obligation among team members, thus enhancing their career commitment.

Employees with high career commitment are more likely to improve workplace wellbeing than those with low career commitment. There are at least two reasons for this. First, employees with a high level of career commitment are likely to identify with their professional goals and feel emotionally attached to them; therefore, they should find what they do to be meaningful and be able connect to their work. Performing work that is perceived as meaningful, in turn, should enhance employees' workplace wellbeing [52]. Second, affectively committed employees should be more satisfied with their job relative to their less committed counterparts [53]; since job satisfaction is an important element of employee wellbeing, we can expect committed employees to have more positive psychological and emotional experiences. In conclusion, career commitment should be a predictor of workplace wellbeing.

Empirically, there is also evidence for linking (a) supervisory career support to career commitment and (b) career commitment to workplace wellbeing. On one hand, supervisory career mentoring [29] and career supports [54] have been found to be positively associated with career commitment. In addition, Son and Kim [55] believe that organizational career growth can effectively enhance employees' commitment to their career. On the other hand, studies have shown that career commitment is associated with variables that are conceptually close to workplace wellbeing, such as subjective career success [56], job sat-

isfaction [57], and subjective wellbeing [25]. In addition, a recent survey of employees in the public service sector showed affective commitment to relate positively to employee wellbeing [58].

As shown above through theoretical explanations and empirical arguments, it is reasonable to assume that employees who perceive a high level of career support from their supervisors are likely to be committed to their career. Committed employees, in turn, will improve their own workplace wellbeing. Taken together, these predictions suggest a previously untested model in which the impact of supervisory career support on workplace wellbeing is transmitted through career commitment. In other words, career commitment may be the more proximal precursor to workplace wellbeing. Accordingly, we suggest that:

**Hypothesis 2.** *Career commitment plays a mediating role between supervisory career support and workplace wellbeing.*

*2.3. The Moderating Effect of FWSS*

FWSS refers to the ease of construction and clarity of an individual's hoped-for work-based identity [28]. This concept of FWSS establishes an important connection between self-concept and future work behavior [59], which is conceptually similar to career insight. Parke et al. [60] believed that salient future work selves would motivate employees to adopt professional behaviors in line with their future image. Future-oriented identities enable the individual to perceive the congruity or incongruity between the present reality and the expected future and shifts the individual's attention from emotional needs to career development needs [28]. As FWSS is an intrinsic incentive resource [61], employees with high FWSS will enhance their sense of career calling [62]. According to the theory of career motivation [26], the consistency or matching degree between individuals and situations is likely to affect individual career decisions and behaviors. The current study, hence, expected the impact of supervisory career support on career commitment to be moderated by FWSS.

Leadership literature holds that a leader's influence on their subordinates can be neutralized by certain individual characteristic variables [63]. For example, general self-efficacy can replace the effect of ethical leadership on employees' work engagement [6]. According to Strauss et al. [28], individuals with low FWSS are more susceptible to external influence due to a less clear understanding of their future work self. That is, such individuals are more likely to accept their supervisors' directions when they guide them toward achieving their career goals, and consequently, their career commitment is enhanced. In contrast, individuals with high FWSS clearly understand their career goals and where they will go [28]. Individuals with high FWSS tend to believe that they cope well with challenges and are more committed [61]. Such traits of individuals are likely to mitigate the career commitment of employees with high FWSS compared to those with low FWSS. Therefore, we propose that:

**Hypothesis 3.** *FWSS moderates the relationship between supervisory career support and career commitment. Specifically, the positive relationship between supervisory career support and career commitment is stronger for employees with low FWSS than with high FWSS.*

When career commitment is conceptually framed as the intensity of an individual's motivation to work in a chosen professional role, the theory of career motivation can be used as a framework to examine the antecedents and consequences of career commitment [30]. The hypothesized model of this study is shown in Figure 1. As a positive situational condition, we expect that supervisory career support can increase employees' career commitment and in turn enhance their workplace wellbeing. In addition, as an individual characteristic, FWSS reflects the individual's clear understanding of their career goals. Since the matching of individual characteristics and situational conditions will affect individual career decisions [26], we assume that the positive effect of supervisory career support on career commitment will be influenced by FWSS. From the perspective

of career motivation, we can obtain a coherent picture of the joint effect, in which situational and individual characteristics together predict employees' career commitment and workplace wellbeing.

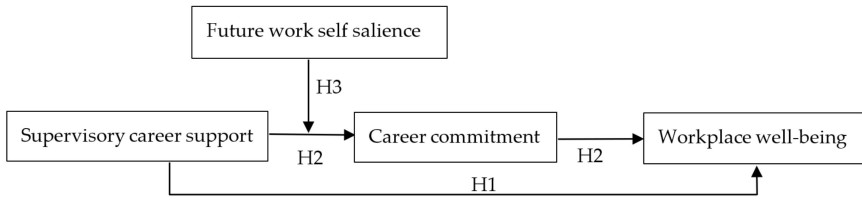

**Figure 1.** Hypothesized model.

### 3. Methods

#### 3.1. Sample and Procedure

We tested our hypotheses using a sample of 213 full-time healthcare workers from three public hospitals located in Southern China. To reduce the common method bias, we conducted a two-wave survey with a one-month time lag, and the data of this study were collected from September to November 2020. At Time 1, participants were asked to report supervisory career support, FWSS, and demographic information (i.e., gender, age, education, and tenure). One month later (Time 2), participants were asked to report their career commitment and workplace wellbeing.

In terms of the first survey (Time 1), we sent questionnaires to 300 healthcare workers with the assistance of human resource departments. In total, 265 participants completed the questionnaires, yielding an 88.33% response rate. One month later (Time 2), 265 questionnaires were distributed to participants who had completed the survey in the first phase, of whom 213 participants completed the survey. Of these healthcare workers (N = 213), 56.34% were doctors and 43.66% were nurses. In total, 40.38% were male, 10.33% under 20 years old, 58.69% between 21 and 25 years old, 23% between 26 and 30 years old, 4.69% between 31 and 35 years old, and 3.29% were 36 and above. With regard to education, 1.41% of the participants had an associate degree, 11.27% a bachelor's degree, 86.38% a master's degree, and 0.94% a doctoral degree. In terms of tenure, 64.32% were employed for 1 to 3 years, 24.41% for 4 to 6 years, 5.16% for 7 to 9 years, and 6.10% had a tenure of at least 10 years.

#### 3.2. Measures

All measurement items were selected from mature scales to ensure validity and reliability. We conducted rigorous two-way translation of the original English measures to ensure translation quality. In addition, to ensure that all items are suitable and applicable to the research context, some minor modifications were made following suggestions from three professors in a relevant research field and five healthcare workers. All variables were measured using a Likert seven-point scale (1 = strongly disagree, 7 = strongly agree).

Supervisory career support was measured using the nine-item scale developed by Greenhaus et al. [24]. One sample: "My supervisor takes the time to learn about my career goals and aspirations". Cronbach's alpha for this scale in this study was 0.965.

Career commitment was measured using the seven-item scale used by Suddaby et al. [64]. One sample: "I am very happy that I chose my current career". Cronbach's alpha for this scale in this study was 0.937.

Future work self-salience was measured using five items adapted from Strauss et al. [28]. One sample: "I can easily imagine my future work self". Cronbach's alpha for this scale in this study was 0.908.

Workplace wellbeing was measured using the six-item scale developed by Zheng et al. [37]. One sample: "I find real enjoyment in my work". Cronbach's alpha for this scale in this study was 0.945.

Control variables. Based on previous studies on employee wellbeing, we controlled for gender, age, education, and organizational tenure. (i) Gender (1 = 'male', 2 = 'female'),

(ii) age (18 = '20 years old or below', 23 = 'within 21–25 years old', 28 = 'within 26–30 years old', 33 = 'within 31–35 years old', and 38 = '36 years old or above'), (iii) education (1 = 'associate degree', 2 = 'bachelor's degree', 3 = 'master's degree', and 4 = 'doctoral degree'), (iv) tenure (2 = 'between 1 and 3 years', 5 = 'between 4 and 6 years', 8 = 'between 7 and 9 years', and 11 = '10 years or above').

## 4. Results

### 4.1. Confirmatory Factor Analysis (CFA)

SPSS 23.0 and Mplus 7.4 were used for statistical analysis in this study. We conducted CFA to statistically distinguish these four key variables in our model (see Table 1). A CFA of this four-factor base model yielded fit indexes within an acceptable range ($\chi^2$/df = 2.91; CFI = 0.91; TLI = 0.90; RMSEA = 0.06; SRMR = 0.09), indicating that these constructs are empirically distinct.

**Table 1.** CFA of measurement models.

| Model | $\chi^2$ | df | $\chi^2$/df | CFI | TLI | SRMR | RMSEA |
|---|---|---|---|---|---|---|---|
| Four-factor model | 917.51 | 315 | 2.91 | 0.91 | 0.90 | 0.06 | 0.09 |
| Three-factor model | 1545.55 | 318 | 4.86 | 0.81 | 0.79 | 0.14 | 0.10 |
| Two-factor model | 2326.77 | 320 | 7.27 | 0.70 | 0.66 | 0.13 | 0.17 |
| One-factor model | 2654.07 | 321 | 8.27 | 0.64 | 0.61 | 0.13 | 0.19 |

Note: The four-factor model includes supervisory career support (SCS), career commitment (CC), future work self-salience (FWSS), and workplace wellbeing (WWB); for the three-factor model, SCS and CC are combined into one factor; for the two-factor model, CC and FWSS are combined into one factor, and so are SCS and WWB; for the one-factor model, SCS, CC, FWSS, WWB are combined into one factor.

We further conducted a convergent validity test by calculating average variance extracted (AVE) and composite reliability (CR). The results in Table 2 show that the factor loadings of each item in the scale were more than 0.6. The AVEs of the scales of supervisory career support, career commitment, future work self-salience, and workplace wellbeing were 0.754, 0.684, 0.703, and 0.743, respectively, all of which are greater than 0.6. CRs were 0.965, 0.937, 0.921, and 0.945, respectively. In conclusion, the data in this study have good convergent validity.

**Table 2.** Item loadings, AVE, and CR.

| Construct | Items | Loading | AVE | CR | Construct | Items | Loading | AVE | CR |
|---|---|---|---|---|---|---|---|---|---|
| Supervisory Career Support | SCS1 | 0.861 | 0.754 | 0.965 | Career Commitment | CC1 | 0.865 | 0.684 | 0.937 |
| | SCS2 | 0.874 | | | | CC2 | 0.915 | | |
| | SCS3 | 0.861 | | | | CC3 | 0.913 | | |
| | SCS4 | 0.832 | | | | CC4 | 0.853 | | |
| | SCS5 | 0.891 | | | | CC5 | 0.825 | | |
| | SCS6 | 0.91 | | | | CC6 | 0.737 | | |
| | SCS7 | 0.848 | | | | CC7 | 0.644 | | |
| | SCS8 | 0.883 | | | | | | | |
| | SCS9 | 0.855 | | | | WWB1 | 0.813 | | |
| Future Work Self-Salience | FWSS1 | 0.612 | 0.703 | 0.921 | Workplace Wellbeing | WWB2 | 0.871 | 0.743 | 0.945 |
| | FWSS2 | 0.896 | | | | WWB3 | 0.814 | | |
| | FWSS3 | 0.841 | | | | WWB4 | 0.895 | | |
| | FWSS4 | 0.927 | | | | WWB5 | 0.848 | | |
| | FWSS5 | 0.877 | | | | WWB6 | 0.926 | | |

### 4.2. Descriptive Statistics and Correlations

Table 3 shows the mean value, standard deviation, and correlation coefficient of the variables. As expected, supervisory career support positively correlated with career commitment and workplace wellbeing ($p < 0.01$), and career commitment positively correlated

with workplace wellbeing ($p < 0.01$). This provides preliminary support for the hypothesis of this study.

**Table 3.** Means, standard deviations, and correlations.

| | M | SD | 1 | 2 | 3 | 4 | 5 | 6 | 7 | 8 |
|---|---|---|---|---|---|---|---|---|---|---|
| 1.Gender | 1.600 | 0.492 | N/A | | | | | | | |
| 2.Age | 24.600 | 4.238 | −0.051 | N/A | | | | | | |
| 3.Education | 2.870 | 0.402 | −0.079 | −0.029 | N/A | | | | | |
| 4.Tenure | 3.590 | 2.549 | 0.041 | 0.655 ** | −0.03 | N/A | | | | |
| 5.SCS | 4.642 | 1.330 | −0.008 | −0.249 ** | −0.058 | −0.176 * | **0.965** | | | |
| 6.CC | 4.850 | 1.182 | 0.118 | −0.135 * | −0.171 * | −0.071 | 0.679 ** | **0.937** | | |
| 7.FWSS | 4.494 | 1.153 | 0.014 | 0.021 | −0.061 | 0.034 | 0.257 ** | 0.406 ** | **0.908** | |
| 8.WWB | 4.565 | 1.214 | 0.053 | −0.167 * | −0.203 ** | −0.097 | 0.605 ** | 0.678 ** | 0.331 ** | **0.945** |

Note: ** $p < 0.01$, * $p < 0.05$. SCS = supervisory career support, CC = career commitment, FWSS = future work self-salience, WWB = workplace wellbeing. Cronbach's alpha coefficient is shown in bold on the diagonal.

### 4.3. Hypotheses Testing

Following calculation of the variance inflation factor to test the collinearity of variables, the results show that all values are less than 3. Therefore, there is no serious collinearity among the variables in this study. To test Hypothesis 1, we first input the control variables (Model 1 in Table 4) and the main effect of supervisory career support (Model 2). After considering the control variables, we found that supervisory career support was positively related to workplace wellbeing ($\beta = 0.539$, $p < 0.01$). Therefore, Hypothesis 1 is supported.

**Table 4.** Results of regression analysis.

| | WWB | | | | | CC | | |
|---|---|---|---|---|---|---|---|---|
| | M1 | M2 | M3 | M4 | M5 | M6 | M7 | M8 |
| Gender | 0.067 | 0.104 | −0.114 | −0.084 | 0.232 | 0.273 * | 0.265 * | 0.276 * |
| Age | −0.052 * | −0.011 | −0.020 | −0.015 | −0.040 | 0.005 | 0.001 | 0.000 |
| Education | −0.620 ** | −0.500 ** | −0.238 * | −0.256 | −0.490 * | −0.355 * | −0.326 * | −0.342 * |
| Tenure | 0.007 | 0.012 | 0.002 | 0.004 | 0.006 | 0.013 | 0.009 | 0.008 |
| SCS | | 0.539 ** | | 0.122 * | | 0.606 ** | 0.547 ** | 0.543 ** |
| CC | | | 0.781 ** | 0.688 ** | | | | |
| FWSS | | | | | | | 0.245 ** | 0.221 ** |
| SCS*FWSS | | | | | | | | −0.058 * |
| R2 | 0.072 | 0.397 | 0.616 | 0.625 | 0.058 | 0.492 | 0.545 | 0.555 |
| △R2 | | 0.325 | 0.544 | 0.553 | | 0.434 | 0.487 | 0.497 |
| F | 4.032 ** | 27.312 ** | 66.461 ** | 57.266 ** | 3.217 * | 40.161 ** | 41.126 ** | 36.557 ** |

Note: N = 213. ** $p < 0.01$, * $p < 0.05$. SCS = supervisory career support, CC = career commitment, FWSS = future work self-salience, WWB = workplace wellbeing.

To test Hypothesis 2, we adopted a combination of hierarchical regression analysis and bootstrap analysis. According to the method of Baron and Kenny [65], supervisory career support is positively related to career commitment ($\beta = 0.606$, $p < 0.01$), as shown in M6 of Table 4. Meanwhile, in M3, career commitment is positively associated with workplace wellbeing ($\beta = 0.781$, $p < 0.01$). The results of M4 further indicate that career commitment was positively related to workplace wellbeing ($\beta = 0.688$, $p < 0.01$) when both supervisory career support and career commitment were regressed on workplace wellbeing. Based on these results, Hypothesis 2 is preliminary supported.

In order to further verify the mediating effect of career commitment, we adopted a bootstrapping analysis (see Table 5). The results showed that the direct and indirect effects of supervisory career support on workplace wellbeing through career commitment were significant at 95%. The two confidence intervals are not zero and support the mediating effect of career commitment. Therefore, Hypothesis 2 is supported.

**Table 5.** Bootstrapping analysis results of the mediation effect in career commitment.

| Effect Categories | Effects | SE | 95% LLCI | 95% ULCI |
|---|---|---|---|---|
| Direct effect | 0.131 * | 0.053 | 0.265 | 0.236 |
| Indirect effect | 0.421 * | 0.057 | 0.319 | 0.541 |

Note: Resampling times = 5000. * $p < 0.05$.

To test Hypothesis 3, we first entered the control variables (Model 5), then the main effect of supervisory career support (Model 7) followed by the two-way interaction of supervisory career support and FWSS (Model 8). The results of Model 7 (see Table 4) show that, after taking control variables into consideration, supervisory career support is positively related to workplace wellbeing (β = 0.547, $p < 0.01$). As shown in Model 8, the interaction term between supervisory career support and FWSS is negative and significant (β = −0.058, $p < 0.05$). Accordingly, Hypothesis 3 is preliminarily supported.

To further explain the interaction term, we drew an interaction diagram [66]. As shown in Figure 2, the resulting pattern is consistent with our prediction. The relationship between supervisory career support and career commitment was slightly stronger when FWSS was below average (Mean − 1SD) (β = 0.643, $p < 0.01$) than when it was above average (Mean + 1SD) (0.4851, $p < 0.01$). Notably, the moderating effect of FWSS was weakly significant. Accordingly, Hypothesis 3 is generally supported.

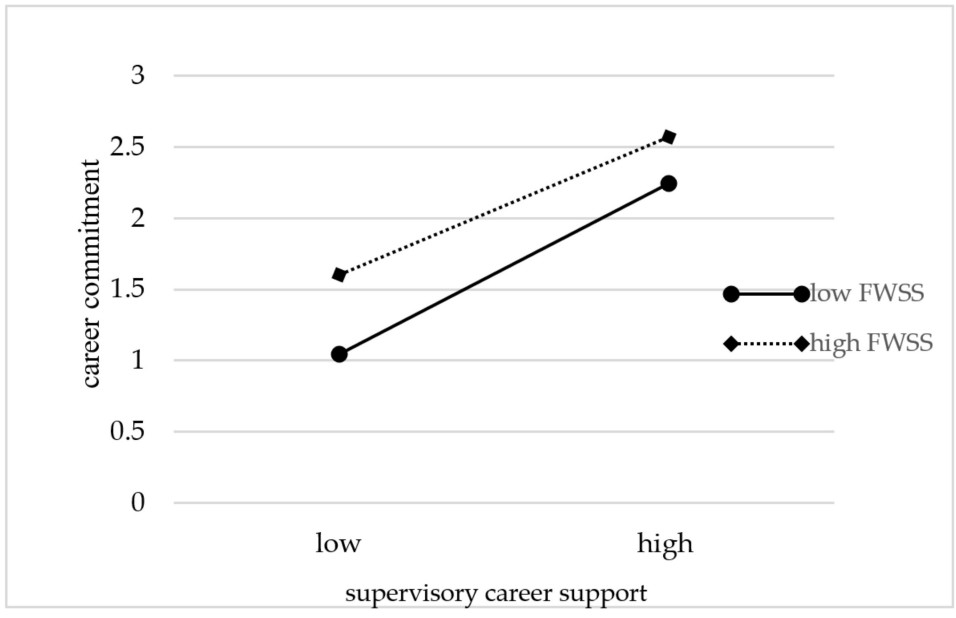

**Figure 2.** Moderating effect of future work self-salience on the relationship between supervisory career support and career commitment.

## 5. Discussion

### 5.1. Theoretical Implications

This study has made several contributions to the related literature. First, we expanded the research on healthcare workers' wellbeing by finding that supervisory career support can be added as a predictor to the list of existing support variables related to workplace wellbeing. The result broadens our understanding of workplace wellbeing from the perspective of career support. In addition, although past studies have established associations between supporting variables and healthcare workers' wellbeing [19,20], this study explains why this may be the case by drawing attention to career commitment as a linking mechanism. Prior study has suggested that supervisory support can promote employees' career management [67]. The results of this study suggest that employees' sense of identity and emotional attachment to their career help to explain why they have positive emotional

experiences at work when they think their career needs are supported. Second, this study contributes to the career decisions and behaviors literature by proving that FWSS can weaken the positive effect of supervisory career support on career commitment. The concept of FWSS represents an individual's hoped-for self in the future [59], which is generally viewed as a positive construct [61]. Employees with high FWSS have a clear picture of their future selves and pay more attention to their career development [28]. However, as shown in this study, when career development opportunities do not match one's career goals, high-FWSS employees may leave their position even if they have a favorable working environment. The theory of career motivation suggests that the effectiveness of career decisions and behaviors depends on the consistency of individual and situational characteristics [26]. The findings of this study expand the theory by demonstrating that individuals with high FWSS tend to be more committed and are less susceptible to supervisory career support.

*5.2. Practical Implications*

This study also provides considerable implications for managers and human resource development practitioners in several ways. First, as indicated by the results of our study, supervisory career support practices tend to encourage commitment of healthcare workers to their career, which in turn helps to enhance their workplace wellbeing. Thus, by promoting supportive leaders and training existing leaders on the traits of career support, organizations will be able to have a team of more committed healthcare workers, hence achieving social sustainability. For instance, in order to identify supportive traits in potential candidates, written tests, structured interviews with a focus on team building, and staff career development should be conducted beyond merely relying on basic background checks [68]. Additionally, diversified career development programs (e.g., formal or informal mentoring programs and career counseling programs) can be provided to support healthcare workers' career goals. It is believed that career growth opportunities can help employees to actively pursue their career goals in their current organizations. Second, this study also presents the substantial implications of FWSS by identifying its role as a moderator in the relationship between supervisory career support and career commitment. Managers providing career support cannot use the same methods to deal with both individuals with low FWSS and high FWSS to extract career commitment from them, as high-FWSS individuals are more resourceful and are thus negatively influenced by extensive directions. Therefore, their career commitment is more easily affected, and an empowerment climate within the organization needs to be built in order to encourage them to stay committed to their company. In contrast, low-FWSS individuals are more susceptible to external influence due to a less clear understanding of their future work self [27]; thus, direction provided by their leaders would enhance their career commitment level. By providing healthcare workers with role clarity through career guidance, challenging tasks, and performance feedback [33] to individuals with low-FWSS, supervisory career support can positively affect their commitment level.

*5.3. Limitations and Future Directions*

Some limitations of this study should be noted. First, although our time-lagged research design can reduce the common method bias, we cannot definitively say that the direction of causality is deterministic. Future studies could use longitudinal research designs to better rule out the possibility of an inverse relationship between career commitment and workplace wellbeing. Second, while this study focuses on how supervisory career support motivates career commitment and workplace wellbeing, it may be other situational characteristics (e.g., leadership style, job design) which influence an individual's career decisions and behaviors. Therefore, the next important step in the study of workplace wellbeing is to explore other situational conditions from different theoretical perspectives. Finally, although the moderating role of FWSS was explored in this study, we did not examine how individual characteristics directly affect career motivation. Therefore, the next important step in the study of career motivation research is to further explore individual factors.

## 6. Conclusions

Based on the theory of career motivation, this study explores the joint effects of supervisory career support and FWSS on career commitment and workplace wellbeing. The results show that career commitment plays a mediating role between supervisory career support and workplace wellbeing. In addition, the effect of supervisory career support on career commitment is stronger among employees with low FWSS than those with high FWSS. Supervisory career support can foster employees' wellbeing at the workplace through building their career commitment and may serve as a potential substitute for a lack of FWSS among employees. Given the findings of this research, we hope that this study will serve as a catalyst for a deeper understanding of career motivation and workplace wellbeing.

**Author Contributions:** Conceptualization, G.X. and H.W.; methodology G.X., Z.L., and H.W.; software, G.X., and Z.L.; validation G.X. and Z.L.; formal analysis, G.X. and Z.L.; investigation, G.X.; resources, G.X. and Z.L.; data curation, G.X.; writing—original draft preparation, G.X. and Z.L.; writing—review and editing, G.X., Z.L., and H.W.; supervision, G.X. and H.W.; funding acquisition, H.W. All authors have read and agreed to the published version of the manuscript.

**Funding:** This research was funded by National Natural Science foundation of China (71872066), General projects of Humanities and Social Sciences of the Ministry of Education (Y9180140), Guang Dong province Natural Science Foundation of China (2020A151501943), and Special project of philosophy and social science planning in Guangdong Province (GD20SQ32).

**Institutional Review Board Statement:** Not applicable.

**Informed Consent Statement:** Not applicable.

**Data Availability Statement:** The datasets used and analyzed in the current study are available from the corresponding author upon reasonable request.

**Acknowledgments:** We would like to thank all the healthcare workers who supported this study and participated in it.

**Conflicts of Interest:** None of the authors have any conflict of interest to report.

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
