# Peer review of "Supervisory Career Support and Workplace Wellbeing in Chinese Healthcare Workers: The Mediating Role of Career Commitment and the Moderating Role of Future Work Self-Salience"

_sustainability, doi:10.3390/su13105572_

Round 1
Reviewer 1 Report
The first introduction is difficult to be understand because some terms and theories are cited but not yet defined.
For instance, “Supervisory career support” is cited several times before to be defined (line 124); the same for the “theory of career motivation”, which should be also better defined.
216 FWSS is not completely described: “salience” is lacking.
298 Cronbach’s alpha coefficients do not seem in bold.
About hypothesis 3, the difference between low and high FWSS does not look strong: an effect size index should be calculated and described.
The “Data analysis” section is lacking.
Author Response
Responses to Reviewer #1:
Thank you for offering constructive suggestions for improving the manuscript. We have revised the paper following your suggestions. All changes are highlighted in yellow in the revision. Here are our responses to your comments:
- The first introduction is difficult to be understand because some terms and theories are cited but not yet defined.For instance, “Supervisory career support” is cited several times before to be defined (line 124); the same for the “theory of career motivation”, which should be also better defined.
Response:
Many thanks for your insightful comments. We have rewritten the introduction and addressed the questions you raised. We have added explanations of some terms (eg., Supervisory career support, line 57-58 in revision; Future work self salience, line 76-77 in revision)and theories. In addition, we have rewritten the section of theoretical backgrounds, and further to explain “theory of career motivation” (line 117-131 in revision). For example, we wrote:
Based on theory of career motivation, we posit that when healthcare workers are gained career support from their supervisors, which signals that supervisors care about their professional needs and development, and enhance their workplace well-being. Furthermore, supervisory career support reflects positive feedback to team members to a certain extent, and can promote the establishment of emotional connections individual and their career. When career commitment is conceptually translated into the intensity of an individual’s motivation to work in a chosen career role, career motivation can be used as a framework to examine how supervisory career support affects an employee’s workplace well-being. In addition, the effect of situational characteristics on career decision-making and career behavior was influenced by individual characteristics. Therefore, the effect of supervisory career support on career commitment may be influenced by FWSS, which represents the ease of construction and clarity of an individual’s hoped-for work-based identity. From a career motivation perspective, we can achieve a coherent picture of the positive influences of supervisory support behavior on healthcare workers.
- 216 FWSS is not completely described: “salience” is lacking.
Response:
Many thanks for your kind suggestion. We have modified the definition of “FWSS” (line 231-232 in revision). For example, we wrote: ‘FWSS refers to the ease of construction and clarity of an individual’s hoped-for work-based identity’.
- 298 Cronbach’s alpha coefficients do not seem in bold.
Response:
Many thanks for your kind suggestion. We have roughened Cronbach’s alpha coefficients in table 3.
- About hypothesis 3, the difference between low and high FWSS does not look strong: an effect size index should be calculated and described.
Response:
Thank you for your guidance. We agree that this effect is relatively weak and give a supplementary explanation. For example, we wrote:
The relationship between supervisory career support and career commitment is slightly stronger when FWSS was below average (Mean - 1 SD) (β=0.643, P<0.01)than when it was above average (Mean + 1 SD)(0.4851, p<0.01). Notably, the moderating effect of FWSS was weakly significant. Accordingly, hypothesis 3 is generally supported. (lines 393-397 in the revision).
- The “Data analysis” section is lacking.
Response:
Many thanks for valuable suggestion. We have added Confirmatory factor analysis (CFA), a CFA of this four-factor base model yielded fit indexes within an acceptable range (χ2/df = 2.91; CFI = 0.91; TLI = 0.90; RMSEA = 0.06; SRMR = 0.09), indicating that these constructs are distinct empirically. We have also provided convergent validity test with item loadings, AVE, and CR. (lines 329-347 in the revision).
Again many thanks for your time and insightful comments and advice, which have helped us strengthen the manuscript.
Reviewer 2 Report
The article is devoted to the actual problem on the relationship between supervisory career support and employee well-being at the healthcare organizations. The authors investigated the mediating role of career commitment and moderating effect of future work self salience (FWSS). And I agree with authors that the significance of the problem on healthcare workers’ well-being has especially increased in the current period of the pandemic.
In my opinion, in general, the article is well written, logically structured and meets the main criteria for scientific publications. However, I have some questions and recommendations to improve the article and provide a more complete presentation of the study.
- The title of the article does not reflect one of the central variables of the research (workplace well-being), as well as the specifics of the sample (Chinese healthcare workers). I propose to modify the title of the article, and I suggestpossible options:
- Supervisory career support and workplace well-being in Chinese healthcare workers: The mediating role of career commitment and the moderating role of future work self salience (full, but too long title)
- Supervisory career support and workplace well-being in Chinese healthcare workers: Search for relationship mechanisms
- ... other options are possible, but I consider it necessary to include the above terms
- It is also necessary to include ‘workplace well-being’ and ‘healthcare workers’ in the list of keywords
- The authors substantiates and presents research hypotheses in great detail, however, I propose to move the Figure 1 and the text on lines 104-122 to the end of the Section 2 after describing all hypotheses.
- Regarding the study design, I don't understand why the survey consisted of two waves? Why is a lag of one month needed if there were no interventions and the dependent variables were not measured at the beginning and at the end of this period.This point needs to be clearly explained, because the authors’ mention that “our time-lagged research design can produce stronger causal inferences …”(lines 386-387), in our opinion, does not make sense.
- At the beginning of the article, the authors write: “After the COVID-19 crisis outbreak, healthcare professionals, like doctors and nurses, are facing unprecedented challenges, exhaustion due to heavy workloads, the fear of becoming infected and infecting others in this dangerous situation [11]. This also has great influence on their career decisions and well-being at the workplace during pandemic threat …”(lines 32-36), but they do not indicate in what time the study was conducted, so it is not clear how it is related to the problems of the pandemic.
- The description of the sample must be supplemented with positions of the surveyed health workers: the percentage of doctors, nurses, etc. must be indicated.
- It is necessary to clarify why such quantitative variables as age and tenure were coded. If this is due to the distribution of these variables, this fact must be indicated and a special procedure should be used to normalize the data.
- I agree that there is an effect of the interaction of independent variables (supervisory career support and future work self salience) on the dependent variable (career commitment), as can be seen in Figure 2. However, this effect is very weak and it should be indicated in the text of the article.
Thus, I believe that this study has scientific and practical significance, but the article needs to be corrected and clarified.
Author Response
Responses to Reviewer #2:
Thank you for offering constructive suggestions for improving the manuscript. We have revised the paper following your suggestions. All changes are highlighted in yellow in the revision. Here are our responses to your comments:
(1) The title of the article does not reflect one of the central variables of the research (workplace well-being), as well as the specifics of the sample (Chinese healthcare workers). I propose to modify the title of the article, and I suggest possible options:
- Supervisory career support and workplace well-being in Chinese healthcare workers: The mediating role of career commitment and the moderating role of future work self salience (full, but too long title)
- Supervisory career support and workplace well-being in Chinese healthcare workers: Search for relationship mechanisms
- ... other options are possible, but I consider it necessary to include the above terms
Response:
We really appreciate for your constructive comment. As suggested, We have decided to modify the title of the article to ‘Supervisory career support and workplace well-being in Chinese healthcare workers: The mediating role of career commitment and the moderating role of future work self salience’. Thanks
- It is also necessary to include ‘workplace well-being’ and ‘healthcare workers’ in the list of keywords.
Response:
Thank you for the valuable comment. We have added the keywords of ‘workplace well-being’ and ‘healthcare workers’.
- The authors substantiates and presents research hypotheses in great detail, however, I propose to move the Figure 1 and the text on lines 104-122 to the end of the Section 2 after describing all hypotheses.
Response:
Thank you for the valuable comment. We have moved the Figure 1 and the text on lines 104-122 to the end of the Section 2 (lines 260-279 in the revision). Thanks.
- Regarding the study design, I don’t understand why the survey consisted of two waves? Why is a lag of one month needed if there were no interventions and the dependent variables were not measured at the beginning and at the end of this period.This point needs to be clearly explained, because the authors’ mention that “our time-lagged research design can produce stronger causal inferences …”(lines 386-387), in our opinion, does not make sense.
Response:
Many thanks for your insightful comments. First, the data collected in this study was single-sourced and self-reported, which may raise common method bias issues. Second, since there’s item overlapping between career commitment and workplace well-being, in order to avoid the common method bias, we gather data on workplace well-being one month later. In addition, We agree with you that “our time-lagged research design can produce stronger causal inferences …” is not very accurate. Accordingly, we have modified it to “ although our time-lagged research design can reduce common method bias, we cannot say definitively that the direction of causality is deterministic” (lines 452-454 in the revision).
- At the beginning of the article, the authors write: “After the COVID-19 crisis outbreak, healthcare professionals, like doctors and nurses, are facing unprecedented challenges, exhaustion due to heavy workloads, the fear of becoming infected and infecting others in this dangerous situation [11]. This also has great influence on their career decisions and well-being at the workplace during pandemic threat …”(lines 32-36), but they do not indicate in what time the study was conducted, so it is not clear how it is related to the problems of the pandemic.
Response:
Many thanks for your insightful comments. This study was conducted after the COVID-19 crisis outbreak, and the data of this study was collected from September to November 2020. we have add data collection time on the section of Methods (lines 284-285 in the revision) .
- The description of the sample must be supplemented with positions of the surveyed health workers: the percentage of doctors, nurses, etc. must be indicated.
Response:
Thank you for your insightful suggestions. We have supplemented positions of the surveyed health workers. “Of these healthcare workers (N=213), 56.34% were doctors and 43.66% were nurses” (lines 293-294 in the revision).
- It is necessary to clarify why such quantitative variables as age and tenure were coded. If this is due to the distribution of these variables, this fact must be indicated and a special procedure should be used to normalize the data.
Response:
Thank you for your insightful comment. We agree with you that such quantitative variables age and tenure should be reported specific values directly. Since respondents are generally reluctant to report their specific age and tenure, we coded these variables according to the relevant literature (eg., He et al., 2019; Liu et al., 2017). In addition, we have transformed categorical variables into continuous variables based on interval median. As you could see from Table 3-4 and the text, this transformation does not have a significant impact on the results.
Reference:
He, G.; An, R.; Zhang, F., Cultural Intelligence and Work–Family Conflict: A Moderated Mediation Model Based on Conservation of Resources Theory. International Journal of Environmental Research and Public Health 2019, 16, (13), 2406.
Liu, X.; Huang, Q.; Wang, H.; Liu, S., Employment security and employee organizational citizenship behavior: does an ‘iron rice bowl’ make a difference? International journal of human resource management 2017, 30, (13), 2077-2096.
- I agree that there is an effect of the interaction of independent variables (supervisory career support and future work self salience) on the dependent variable (career commitment), as can be seen in Figure 2. However, this effect is very weak and it should be indicated in the text of the article.
Response:
Thank you for your guidance. We agree that this effect is relatively weak and give a supplementary explanation. “The relationship between supervisory career support and career commitment is slightly stronger when FWSS was below average (Mean - 1 SD) (β=0.643, P<0.01)than when it was above average (Mean + 1 SD)(0.4851, p<0.01). Notably, the moderating effect of FWSS was weakly significant. Accordingly, hypothesis 3 is generally supported” (lines 393-397 in the revision).
.
Again many thanks for your time and insightful comments and advice, which have helped us strengthen the manuscript.
Reviewer 3 Report
Explain here why your study is important? What new information are you providing? As for the literature review, I think the author has done a good job but failed to highlight the theoretical framework of their study well. Also, by stating the previous literature findings the hypothesis cannot be formulated. The author(s) should strengthen the literature review to support the proposed hypotheses. They need to go beyond merely reporting previous findings, author(s) should discuss from their perspective. For example, the Literature review should reflect the author(s) opinions for proposing the study's hypotheses based on the previous findings. Directionality and Number of the hypothesis is recommended to be marked in a model of Research. Please mention Fig. Research Model. Have you conducted a pilot test? Details on the pilot test, the method and place of applying the questionnaires, etc. should be explained in detail. Significance level with t-value following CFA is suggested. (Also, AVE, CCR, and Cronbach’s alpha). Factor analysis (EFA or CFA) performed by variable does not make sense. The results of discriminant validity, reliability and convergent validity test were not presented. Finally, what are the practical implications based on findings of this study? More specific and realistic (substantial) implications are required. It is difficult to recognize difference from the already-preceded research. In addition, even the supplementation is necessary for research limitations and future research.Author Response
Responses to Reviewer #3:
Thank you for offering constructive suggestions for improving the manuscript. We have revised the paper following your suggestions. All changes are highlighted in yellow in the revision. Here are our responses to your comments:
- Explain here why your study is important? What new information are you providing? Response:
Many thanks for your insightful comments. We have rewritten the introduction and addressed the questions you raised.
The purpose of this study is to explore how and when supervisor career support affects workplace well-being of healthcare workers. Specifically, we mainly examined the mediating role of career commitment and the moderating role of future work self silence. The importance of this study is as follows. Practically, as we all known, employee well-being is a particularly important issue for sustainable organizations because it is a key driver of employees’ affective commitment, job performance and organizational citizenship behavior (Darvishmotevali & Ali, 2020; Dipietro et al., 2020; Paul et al., 2019). However, during the COVID-19 crisis outbreak, healthcare professional, like doctors and nurses, their well-being are facing unprecedented challenges, exhaustion due to heavy workloads, the fear of becoming infected and infecting others in this dangerous situation (Liu et al., 2020). Healthcare professionals with a higher level of well-being are more likely to be enthusiastic about functioning and provide patient-centred care (Giesbers et al., 2015). In view of employee well-being is one of the sustainable goals of the organization, it is necessary to investigate how to enhance healthcare workers’ well-being during the COVID-19 pandemic, which is of great significance for organizations to maintain sustainable goals in the face of external crisis (Creese et al., 2021; Salas-Vallina et al., 2020).
Theoretically, first, although the existing research has examined the impact of supervisor support on healthcare workers’ well-being (Seguin, 2019; Sturges et al., 2000), we know relatively little about the effects of career support variables on workplace well-being. This study explored the influence of supervisor support on workplace well-being from the perspective of career development. From the perspective of career support, we give a new idea on how to enhance heathcare workers’ well-being in face of external crisis. Second, in previous work, the researchers focused on the direct impact of supporting variables on employees’ well-being. This study further integrates insights from the career motivation literature to examine the mediation model, which links the supervisory career support and workplace well-being through the mediating mechanism of career commitment, which represents the career-based identities of employees. That is, this study highlights the significance of supervisor support to enhance healthcare workers’ career commitment. Finally, this study further examined the moderating effect of personality on the relationship supervisory career support and career commitment. by examining the moderating role of FWSS, this study aims to prove that the influence of supervisory career support on career commitment is not universal among individuals, but depends on the significance of their expected job-related identity.
Reference:
Agnes, B.; Nicole, G. Team Lavender: Supporting employee well-being during the COVID-19 pandemic. Nursing 2021, 51, 16-19.
Creese, J.; Byrne, J.P.; Conway, E.; Barrett, E.; Prihodova, L.; Humphries, N. “We All Really Need to just Take a Breath": Composite Narratives of Hospital Doctors' Well-Being during the COVID-19 Pandemic. Int. J. Environ. Res. Public Health 2021, 18, 2051.
Darvishmotevali, M.; Ali, F. Job insecurity, subjective well-being and job performance: The moderating role of psychological capital. Int. J. Hosp. Manag. 2020, 87, 1-10.
Dipietro, R.B.; Moreo, A.; Cain, L. Well-being, affective commitment and job satisfaction: influences on turnover intentions in casual dining employees.J. Hosp. Market. Manag. 2020, 29, 139-163.
Giesbers, A.P.M.; Schouteten, R.L.J.; Poutsma, E.; Van der Heijden, B.I.J.M.; Van Achterberg, T. Feedback provision, nurses' well-being and quality improvement: towards a conceptual framework. J. Nurs. Manag. 2015, 23, 682-691.
Liu, Q.; Luo, D.; Haase, J. E.; Guo, Q.; Wang, X.Q.; Liu, S.; Xia, L.; Liu, Z.; Yang, J.; Yang, B.X. The experiences of health-care providers during the COVID-19 crisis in China: A qualitative study. Lancet Glob. Health 2020, 8, e790-e798.
Lopez-Cabarcos, M. A.; Lopez-Carballeira, A.; Ferro-Soto, C. New Ways of Working and Public Healthcare Professionals' Well-Being: The Response to Face the COVID-19 Pandemic. Sustainability 2020, 12, 8087.
Paul, H.; Bamel, U.; Ashta, A.; Stokes, P. Examining an integrative model of resilience, subjective well-being and commitment as predictors of organizational citizenship behaviors. International Journal of Organizational Analysis, 2019, 27, 1274-1297.
Seguin, C. A survey of nurse leaders to explore the relationship between grit and measures of success and well-being. J. Nurs. Adm. 2019, 49, 125-131.
Sturges, J.; Guest, D.; Mackenzie D.K. Who’s in charge? Graduates’ attitudes to and experiences of career management and their relationship with organizational commitment. Eur. J. Work Organ. Psychol. 2000, 9, 351-370.
- As forthe literature review, I think the author has done a good job but failed to highlight the theoretical framework of their study well.
Response:
Many thanks for your insightful comments. We have rewritten the section of theoretical backgrounds, and further to highlight the theoretical framework of this study (line 117-131 in revision). For example, we wrote:
Based on theory of career motivation, we posit that when healthcare workers are gained career support from their supervisors, which signals that supervisors care about their professional needs and development, and enhance their workplace well-being. Furthermore, supervisory career support reflects positive feedback to team members to a certain extent, and can promote the establishment of emotional connections individual and their career. When career commitment is conceptually translated into the intensity of an individual’s motivation to work in a chosen career role, career motivation can be used as a framework to examine how supervisory career support affects an employee’s workplace well-being. In addition, the effect of situational characteristics on career decision-making and career behavior was influenced by individual characteristics. Therefore, the effect of supervisory career support on career commitment may be influenced by FWSS, which represents the ease of construction and clarity of an individual’s hoped-for work-based identity. From a career motivation perspective, we can achieve a coherent picture of the positive influences of supervisory support behavior on healthcare workers.
- Also, by stating the previous literature findings the hypothesis cannot be formulated. The author(s) should strengthen the literature review to support the proposed hypotheses. They need to go beyond merely reporting previous findings, author(s) should discuss from their perspective. For example, the Literature review should reflect the author(s) opinions for proposing the study's hypotheses based on the previous findings. Directionality and Number of the hypothesis is recommended to be marked in a model of Research. Please mention Fig. Research Model.
Response:
Many thanks for your insightful comments. We have rewritten the section of “Theoretical backgrounds and hypotheses”. On one hand, we have rewritten Theoretical backgrounds, and further clarified how we can achieve a coherent picture of the positive influences of supervisory support behavior on healthcare workers from a career motivation perspective. On other hand, we have added the reasoning process of research hypotheses from theoretical point.
- Have you conducted a pilot test? Details on the pilot test, the method and place of applying the questionnaires, etc. should be explained in detail.
Response:
Many thanks for your insightful comments. All measurement items were selected from mature scales to ensure validity and reliability. We conducted rigorous two-way translation of the original English measures to ensure translation quality. In addition, to ensure that all the items are suitable and applicable to the research context, some minor modifications were made following suggestions from three professors in a relevant research field and five healthcare workers. All variables were measured by Likert seven-point scale (1 = strongly disagree, 7 = strongly agree). We have rewritten the process of measurement (lines 302-308 in the revision). In addition, we have added the reliability and validity tests. The results shows that the data in this study has good reliability and validity.
- Significance level with t-value following CFA is suggested. (Also, AVE, CCR, and Cronbach’s alpha). Factor analysis (EFA or CFA) performed by variable does not make sense. The results of discriminant validity, reliability and convergent validity test were not presented.
Response:
Many thanks for your kind suggestion. We have added Confirmatory factor analysis (CFA), a CFA of this four-factor base model yielded fit indexes within an acceptable range (χ2 /df = 2.91; CFI = 0.91; TLI = 0.90; RMSEA = 0.06; SRMR = 0.09), indicating that these constructs are distinct empirically. We have also provided convergent validity test with item loadings, AVE, and CR.
- Finally, what are the practical implications based on findings of this study? More specific and realistic (substantial) implications are required. It is difficult to recognize difference from the already-preceded research.
Response:
Many thanks for your insightful comments. We have rewritten the section of “5.2 Practical implications” (lines 428-450 in the revision). For example, we wrote:
Several practical implications can be obtained from the results of this study. First, as healthcare organizations struggle to protect their employees’ psychological health in a very constrained context, this study sheds light on some factors that could contribute to promote healthcare workers’ career commitment and workplace well-being, by emphasizing the importance of career support from supervisors and organizations. Specifically, our research underscores the necessity for organizations to foster their career development policies so that healthcare workers develop positive assessments about their career opportunities. In addition, managers should provide diversified organizational practices (eg., formal or informal mentoring programs and career counseling programs) to support healthcare workers’ career development and career goals, as these interventions motivate employees to commit to their careers, which in turn improves their workplace well-being. It is believed that career growth opportunities can help employees actively pursue their career goals in the current organization. Third, this study also presents substantial implications of FWSS by identifying its role as moderator in the relationship between supervisor career support and career commitment. Managers by playing the role of career support to deal with individuals with low FWSS and high FWSS not on the same pace to extract career commitment from them. As, low FWSS individuals are more susceptible to external influence due to lesser clear understanding of their future work self [27], thus directing them accordingly by leaders would enhance their career commitment. By providing healthcare workers with role clarity, through career guidance, challenging tasks, and performance feedback to individuals with low FWSS, supervisor career support positively affect their their commitment level.
Reference:
Strauss, K.; Griffin, M.A.; Parker, S.K. Future work selves: How salient hoped-for identities motivate proactive career behaviors. J. Appl. Psychol. 2012, 97, 580-598.
- In addition, even the supplementation is necessary for research limitations and future research.
Response:
Thank you for your comment. We have stated limitations and future research in our manuscript (lines 451-464 in the revision).
Again many thanks for your time and insightful comments and advice, which have helped us strengthen the manuscript.
Round 2
Reviewer 3 Report
More specific and realistic (substantial) implications are required.
Author Response
Responses to Reviewer #3:
Thank you for offering constructive suggestions for improving the manuscript. We have revised the paper following your suggestions. All changes are highlighted in yellow in the revision. Here are our responses to your comments:
(1)More specific and realistic (substantial) implications are required.
Response:
Many thanks for your insightful comments. We have rewritten the section of “5.2 Practical implications” (lines 428-453 in the revision). For example, we wrote:
This study also provides considerable implications for managers and human resource development practitioners in several ways. First, as indicated by the results of our study, supervisor career support practices tend to bring out the commitment of healthcare workers to their career, which in turn helps to enhance their workplace well-being. Thus by promoting supportive leaders and training existing leaders the traits of career support, organizations will be able to have more committed healthcare workers hence achieving social sustainability. For instance, in order to identify supportive traits in potential candidates, written tests, structured interviews with focus on team building and staff career development should be conducted beyond merely relying on basic background check (Brody, 2010). Besides, diversified career development programs (eg., formal or informal mentoring programs and career counseling programs) can be provided to support healthcare workers’ career goals. It is believed that career growth opportunities can help employees actively pursue their career goals in the current organization. Second, this study also presents substantial implications of FWSS by identifying its role as moderator in the relationship between supervisor career support and career commitment. Managers by playing the role of career support to deal with individuals with low FWSS and high FWSS not on the same pace to extract career commitment from them. As, high FWSS individuals being more resourceful are negatively influenced by extensive directions and thus their career commitment is affected, so in order to make them committed, empowerment climate within the organization needs to be built. In contrast low FWSS individuals are more susceptible to external influence due to lesser clear understanding of their future work self (Strauss et al., 2012), thus directing them accordingly by leaders would enhance their career commitment level. By providing healthcare workers with role clarity, through career guidance, challenging tasks, and performance feedback (Yang et al., 2018) to individuals with low FWSS, supervisor career support positively affect their their commitment level.
Reference:
Strauss, K.; Griffin, M.A.; Parker, S.K. Future work selves: How salient hoped-for identities motivate proactive career behaviors. J. Appl. Psychol. 2012, 97, 580-598.
Brody, R.G. Beyond the basic background check: hiring the ‘right’ employees. Manage. Res. Rev. 2010, 33, 210-223 .
Yang, F.; Liu, J.; Huang, X.; Qian, J.; Wang, T.; Wang, Z.; Yu, R.P. How supervisory support for career development relates to subordinate work engagement and career outcomes: the moderating role of task proficiency. Hum. Resour. Manag. J. 2018, 28, 496-509.
Again many thanks for your time and insightful comments and advice, which have helped us strengthen the manuscript.